# Large-scale integration of single-cell transcriptomic data captures transitional progenitor states in mouse skeletal muscle regeneration

David W. McKellar[1], Lauren D. Walter[2], Leo T. Song [1], Madhav Mantri [3], Michael F. Z. Wang[3], Iwijn De Vlaminck [1,4✉] & Benjamin D. Cosgrove [1,4✉]

Skeletal muscle repair is driven by the coordinated self-renewal and fusion of myogenic stem and progenitor cells. Single-cell gene expression analyses of myogenesis have been hampered by the poor sampling of rare and transient cell states that are critical for muscle repair, and do not inform the spatial context that is important for myogenic differentiation. Here, we demonstrate how large-scale integration of single-cell and spatial transcriptomic data can overcome these limitations. We created a single-cell transcriptomic dataset of mouse skeletal muscle by integration, consensus annotation, and analysis of 23 newly collected scRNAseq datasets and 88 publicly available single-cell (scRNAseq) and single-nucleus (snRNAseq) RNA-sequencing datasets. The resulting dataset includes more than 365,000 cells and spans a wide range of ages, injury, and repair conditions. Together, these data enabled identification of the predominant cell types in skeletal muscle, and resolved cell subtypes, including endothelial subtypes distinguished by vessel-type of origin, fibro-adipogenic progenitors defined by functional roles, and many distinct immune populations. The representation of different experimental conditions and the depth of transcriptome coverage enabled robust profiling of sparsely expressed genes. We built a densely sampled transcriptomic model of myogenesis, from stem cell quiescence to myofiber maturation, and identified rare, transitional states of progenitor commitment and fusion that are poorly represented in individual datasets. We performed spatial RNA sequencing of mouse muscle at three time points after injury and used the integrated dataset as a reference to achieve a high-resolution, local deconvolution of cell subtypes. We also used the integrated dataset to explore ligand-receptor co-expression patterns and identify dynamic cell-cell interactions in muscle injury response. We provide a public web tool to enable interactive exploration and visualization of the data. Our work supports the utility of large-scale integration of single-cell transcriptomic data as a tool for biological discovery.

[1] Meinig School of Biomedical Engineering, Cornell University, Ithaca, NY 14853, USA. [2] Department of Molecular Biology & Genetics, Cornell University, Ithaca, NY 14853, USA. [3] Department of Computational Biology, Cornell University, Ithaca, NY 14853, USA. [4] These authors contributed equally: Iwijn De Vlaminck, Benjamin D. Cosgrove. ✉email: id93@cornell.edu; bdc68@cornell.edu

Muscle stem cells (MuSCs) are essential for muscle homeostasis and repair. MuSCs are typically quiescent in homeostasis and are activated after muscle damage. Their subsequent proliferation, differentiation, commitment, and fusion replenishes skeletal muscle tissue in a complex, coordinated process[1–3]. MuSCs are a rare cell type, accounting for less than 1% of the cells within skeletal muscle at homeostasis. Even rarer are the cell states quiescent MuSCs transition through during differentiation to myofiber cells. Consequently, MuSCs and muscle progenitor cells (myoblasts and myocytes) are difficult to study in their native tissue context. Conventional strategies to study MuSCs and muscle progenitor cells rely on enrichment by fluorescence-activated cell sorting using a transgenic reporter or prospective isolation markers[4]. These methods however are ill-suited to capture the subtle, continuous cell state transitions which are critical for myogenesis due to a paucity of highly stage-specific cell isolation markers and the rarity of these cells.

Single-cell RNA sequencing (scRNAseq) enables a detailed characterization of cell types and states in complex tissues without the need for targeted cell enrichment[5–8]. Skeletal muscle has been the focus of a number of recent scRNAseq studies, which have aimed to catalog its dynamic and heterogeneous constituent cell types and the progression of myogenic stem and progenitor cell regulation in muscle development and repair[7]. Single-nucleus RNA sequencing (snRNAseq) has been used to capture transcriptomic signatures from mature myofiber nuclei, which are largely lost during cell isolation required for scRNAseq[9–13]. Yet, despite advances in the scale of sc/snRNAseq technologies ($10^3–10^4$ cells per experiment), these methods still poorly sample rare cell types and transient cell states in detail without purification, which can introduce marker bias and technical artefacts[14]. For example, we previously used scRNAseq to study the dynamics of hindlimb skeletal muscle regeneration in adult mice and resolved ~12 muscle-resident cell types from ~35,000 single-cell transcriptomes[15]. However, we observed fewer than 100 committed and fusing myogenic cells even though we sampled key time-points of myogenic differentiation post-injury[15]. Other studies similarly reported an infrequent sampling of committed myogenic progenitors from whole muscle samples[15–17].

To overcome these challenges, we used large-scale integration of single-cell transcriptomics data. We measured ~95,000 single-cell transcriptomes from 23 new samples of regenerating mouse hindlimb muscles in older mice. We then leveraged recent improvements in batch-correction algorithms[18,19] to incorporate 88 publicly available sc/snRNAseq datasets from 18 prior studies in our analysis[9,11,15–17,20–32]. This led to a dataset that included ~365,000 cells/nuclei after quality filtering and allowed us to study the cellular composition and dynamics in response to skeletal muscle injury over a wide range of experimental conditions. The depth of transcriptome coverage achieved by large-scale integration of single-cell transcriptomic data enabled us to robustly characterize rare, short-lived cell states on the myogenic cell differentiation trajectory. We identified transcription factors and surface markers that distinguish committed myoblasts (~5 per sample, on average) and fusing myocytes (~15 per sample, on average), which represent only 0.2 and 0.5% of all cells in the integrated muscle compendium, respectively. We performed spatial RNA sequencing of mouse muscle at three-time points after injury and used the integrated compendium as a reference to achieve a high-resolution, local deconvolution of cell subtypes. Our analysis brings insights into the dynamics of stromal and immune cell colocalization with transient myogenic cell states.

## Results

### Large-scale integration enables a high-resolution view of skeletal muscle. To profile skeletal muscle homeostasis and repair,

we performed scRNAseq on 23 adult mouse skeletal muscle samples using the 10x Chromium v3 platform. In addition to uninjured controls, we induced muscle damage in adult (7 mo) and aged (20 mo) C57BL/6J mice using notexin and collected tibialis anterior muscles at several time points within one week after injury (sample details in Table S1 and Supplementary Data 1). To augment these data, we curated 88 publicly available mouse skeletal muscle sc/snRNAseq datasets that were generated on the 10x Chromium platform (v2, v3, or v3.1) from PanglaoDB[33] and SRA as of January 1, 2021. Below, we refer to each dataset by its citation, using the first name(s) listed in either the publication or deposition (for data not published) and year of release. Together, these comprised 111 individual samples with a total of 503,929 cell barcodes, before quality control and filtering (Figs. 1a and S1, Table S1). These data vary across sex, age (10 days to 30 months of age), chemical injury model (notexin and cardiotoxin), injury-response timepoint (0.5–21 days post-injury [dpi]), and the sample preparation strategy, including whole-muscle dissociations and FACS enrichment of specific cell types (Figs. 1b and S1).

We downloaded and reanalyzed raw sequencing data using a common pipeline (see "Methods"). First, we re-aligned reads to a common reference genome (mm10). After removal of ambient RNA signatures (SoupX[34]), filtering of low-quality cells, and identification of doublets (DoubletFinder[35]), we merged the datasets and performed initial single-cell transcriptomic analyses with Seurat[36]. Because of the range of data sources, experimental conditions, and differences in library preparation, substantial batch effects were evident in dimensionally-reduced visualizations of these data (Figs. S1 and 1c). To remove these batch effects, we integrated the datasets using three recently described approaches: Harmony, Scanorama, and BBKNN[37–39] (Figs. 1d and S2). These three methods were selected based on criteria from previous benchmark studies, including scalability and trajectory conservation[18,19]. After quality control and integration, the resulting dataset consisted of 365,011 cells and nuclei. In a comparison of integration results across the three methods, we found that Harmony excelled based on compute speed and memory criteria and performed best at integrating the heterogeneous datasets based on multiple observations (Figs. S2, S3). First, we compared gene expression signatures to classify the cell subtypes generated after integration and found those from Harmony were the most consistent with the literature and resulted in more frequent and higher expression of key marker genes (e.g., *Myog* in myogenic progenitors). Second, Harmony was also the only method able to resolve the differences between single-cell and single-nucleus datasets across all cell types (Fig. S2). Third, Harmony best maintained local and global structure within the dimensionally reduced space, positioning it best for downstream analyses like pseudotime trajectory inference. Notably, Harmony did not leave some samples in distinct unmixed clusters and yielded a continuous structure for all myogenic cells in the UMAP representation, whereas other integration methods failed in these regards (Figs. 1c and S2).

After Harmony integration, we performed clustering and used canonical marker genes to manually identify cell types for each batch-correction output independently (Fig. S2b). The size of the dataset enabled the identification of subtle cell-type differences. For example, we were able to distinguish the blood vessel type for endothelial cells[20], the fiber type for mature myonuclei[9,11,40], and the differentiation potential for fibro-adipogenic progenitors (FAPs[41]) (Figs. 1d and S3). The gene panels used for cell type classification are shown in Fig. S3. We found that low-resolution cell type labels (lymphocytes, myeloid cells, endothelial cells, FAPs, neural cells, smooth muscle cells, and myogenic cells) were largely consistent for each batch-correction method, but cell

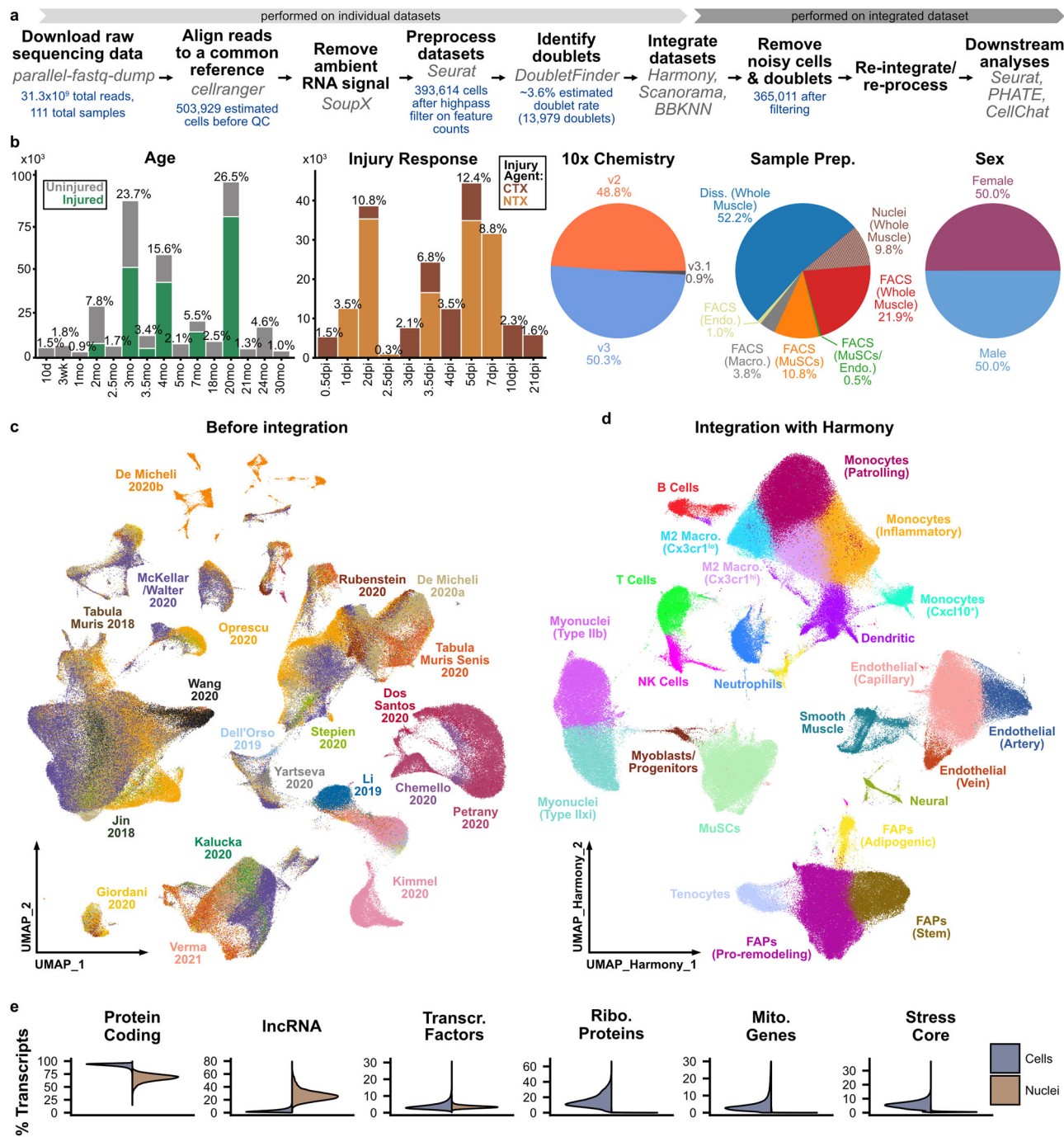

**Fig. 1 Large-scale integration of 111 single-cell and single-nucleus RNAseq samples reveals cell subtypes in skeletal muscle. a** Workflow used for preparation, integration, and analysis of sc/snRNAseq compendium (see "Methods"). **b** Overview of experimental and technical variables across compendium. The percentages shown are calculated with respect to cell number after quality control. Ages in months (mo). Injury by cardiotoxin (CTX) or notexin (NTX). Time-points in days post-injury (dpi). See also Table S1. **c** UMAP representation of the merged datasets after alignment, ambient RNA removal, quality control filtering and doublet removal, but before batch-correction, colored by the dataset source. **d** UMAP representation of integrated compendium after batch-correction with Harmony. Cells are colored by cell type, identified after Harmony integration (Fig. S3). **e** Differential detection of gene biotype sets between single-cell and single-nucleus datasets, including all protein-coding genes, long noncoding RNAs (lncRNAs), transcription factors, cell surface proteins, ribosomal protein subunits, mitochondrial genes, and "core" dissociation-associated stress factors.

subtype labels varied. Variation in gene expression patterns was especially strong within the monocyte and macrophage subtypes, likely reflecting the subtle differences in transcriptional activity in these highly plastic cells. After global clustering, we identified additional cell-subtype-specific marker genes for FAPs, endothelial cells, and myeloid immune cells through differential gene expression analysis (Fig. S4 and Supplementary Data 4–6). We

also performed dimensional reduction with the recently described tool PHATE (Potential of Heat-diffusion for Affinity-based Trajectory Embedding[42]) to visualize similarities between each of the constituent cell subtypes (Fig. S4b) and injury response timepoints (Fig. S4b).

We examined the effects of sample preparation (single-cell or single-nuclei) and 10x Genomics Chromium chemistry (v2, v3, or

v3.1) on the measured transcriptomes (Figs. 1e and S1). We found that the Chromium chemistries provide similar sensitivity and that differences between the datasets were largely driven by sequencing depth (Fig. S1). In contrast, the most substantial differences were observed between the single-cell and single-nuclei preparations. These differences include an increase in intronic and intergenic reads, an increase in noncoding RNA detection, and a decrease in mitochondrial and ribosomal protein transcripts for the single-nuclei, in comparison to the single-cell preparations (Figs. 1e and S5). Consistent with a previous analysis[40], we also observed that single-cell data is enriched for genes associated with dissociation-induced stress (e.g., *Jun*). We further examined whether there were cell type-specific differences in single-cell and single-nucleus data. We identified differentially expressed genes between these assay formats within the most abundant cell types (Figs. S5, S6). We found that protein coding (*Ttn*), mitochondrial (*mt-Atp6*), ribosomal (*Rpl13*), heat-shock (*Hspa1a*), and surface protein (*Ly6e*) genes have more frequent transcript detection in single-cell data, whereas lncRNAs (*Meg3*) are elevated in single-nucleus data, across almost all cell types. In contrast, transcripts encoding transcription factors show minimal differences in most cell types, but some individual transcripts (*Cebpb*, *Ybx1*) are more prone to cell-to-nucleus differences. This suggests that sample preparation imparts a bias in detection frequency and magnitude, regardless of cell type. Many of these effects are likely based on the cellular localization of RNA type. Notably, the single-nucleus data is enriched in myofiber transcripts (e.g., *Myh4*) even in non-myogenic cells, suggesting that cell lysis leads to ambient myofiber-derived accumulation that is not completely removed by SoupX.

**Integrated pseudotime analysis reveals the complete trajectory of native myogenic differentiation.** Previous differentiation trajectory analyses of the myogenic cell lineage have elucidated transcriptional dynamics of in vivo muscle regeneration[15,17] and degeneration[23], or in vitro activation[28]. However, because of the relatively small size of the datasets in these studies, the reported trajectories contained gaps within the continuum of myogenesis and contain very few cells from the most short-lived states of myogenic commitment and fusion. To fill these gaps, we selected all 84,383 myogenic cells to construct a continuous, consensus landscape of myogenesis. Importantly, the size and complexity of the myogenic cells required a scalable workflow that incorporates batch-corrected values. We found that using PHATE to dimensionally-reduce Harmony values generated continuous embeddings which reflected canonical expression patterns of myogenesis (Fig. 2a, e).

Because the first dimension of the Harmony-derived PHATE[42] embedding accounted for 95.6% of the variance, it was possible to use it as a proxy for myogenic differentiation status (Fig. 2b–e). We separated the cells into 25 evenly spaced bins along the differentiation axis to visualize changes in gene expression (Fig. 2b). This binning demonstrates the relative sparsity of intermediate myogenic cell states (Fig. 2c). We computed a Simpson diversity index on sample identifiers in each bin and found that the samples were well-mixed across the embedding (Simpson index near 1), suggesting effective batch-correction with Harmony, but fewer samples were represented in intermediate bins. This sparsity and a slight decline in diversity both suggest that late fusion states are non-redundantly captured across the datasets within the compendium. Canonical marker gene analysis confirmed the dataset captured cells from the major phases of myogenesis, including *Pax7*[hi] quiescence, *Myf5*[hi] activation, *Cdkn1c*[hi] proliferation, *Myog*[hi] commitment, *Mymx*[hi] fusion, and *Acta1*[hi] maturation (Fig. 2d). Bins #1–3 were highly

enriched with sorted then cultured MuSCs[28]. Consistent with previous trajectory analyses of myogenic differentiation[15,17], the proportion of cells captured in intermediate differentiation states (defined as bins #8–18) was very small (0.65% of the compendium). Nonetheless, the large-scale integration of 111 transcriptomic datasets yielded 2,366 committed or fusing cells enabling a detailed analysis of gene expression in these cell populations.

We next explored the continual changes in myogenic differentiation. Recent work has shown that the overall number of genes expressed within a cell narrows during mouse development and human mesodermal lineage specification[43]. To test whether this is also a feature of adult myogenesis, we normalized the number of features in each cell to the sequencing saturation from that sample to account for differences in sequencing depth and found that the number of genes expressed per cell is increased in myogenic progenitor cell states (Fig. 2d). Notably, this burst in transcript diversity marks the exit from quiescence, continues through differentiation, and then is suppressed in mature myofibers. We found that activated (bins 6–7), committed (bins 8–10), and fusing (bins 11–18) myogenic cells were enriched within injured samples, most notably at time points beyond four days post-injury (Fig. 3a). Within the committed and fusing compartments (2,366 total cells), 92 of the 111 samples were represented, with between one and 222 cells captured from each sample (Fig. 3b). This observation again underscores the paucity of these cells within each individual sample and the power of large-scale integration for analyzing rare cell states.

Two classes of genes, surface proteins and transcription factors, are critical for the study of MuSCs and myogenesis[3,4,44]. Surface proteins are used to isolate MuSCs and determine their regenerative potency, and the ordered expression of transcription factors has been used to define the progression of differentiation. Because these genes tend to be lowly expressed[44], we reasoned that our large-scale integrated dataset, which continually spans myogenesis, may be able to identify new markers, especially for transient committed cells. We performed differential gene expression (Wilcoxon Rank Sum test, Methods) between each of the bins along the first PHATE dimension. After filtering (adjusted *p*-value $<10^{-10}$, average $\log_2$-fold-change $> 0.5$), we found 2,031 genes that are differentially expressed across differentiation. We selected surface markers among genes that were enriched in committed and fusing cells (bins 8–18), by cross-comparing against a list of surface proteins generated by the Cell Surface Protein Atlas[45]. Of the 67 resulting genes, *Timp1*, *Clcn5*, *Igfbp3*, *Bst2*, *Gpc1*, *Cd97*, *Lrrn1*, *Megf10*, *Slc29a*, *Jam3*, *Cd164*, *Ncam1*, *Cdh2*, *Fndc5*, *Erbb3*, *Fam171a2*, *Igf2r*, *Ppap2a*, and *Bcam* were specifically and highly expressed in committed and fusing myogenic cells (Fig. 3d and Supplementary Data 3). Notably, we also observed latent expression of surface markers widely used to select MuSCs in committed and fusing cells (*Itgb1*, *Vcam1*, *Cd34*, *Cxcr4*, and *Itga7*). We next filtered the 2,031 differentially expressed genes for transcription factors (MGI GO term "DNA-binding transcription factor activity") that were enriched in committed and fusing cells. Among the 86 resulting genes, we found 12 transcription factors that were highly expressed only during the commitment and fusion stages of myogenesis (*Myog*, *Ctnnb1*, *Purb*, *Zbtb18*, *Mycl*, *Scx*, *Mef2a*, *Hes6*, *E2f8*, *Klf5*, *Id1*, *Tead4*). Overall, these findings demonstrate that large-scale integration of single-cell transcriptomic data can enable detailed gene expression profiling of rare cell states, even for lowly expressed transcripts. The surface markers identified here may be used to deplete more committed cells in MuSC isolation protocols or more selectively enrich transient committed progenitor cells.

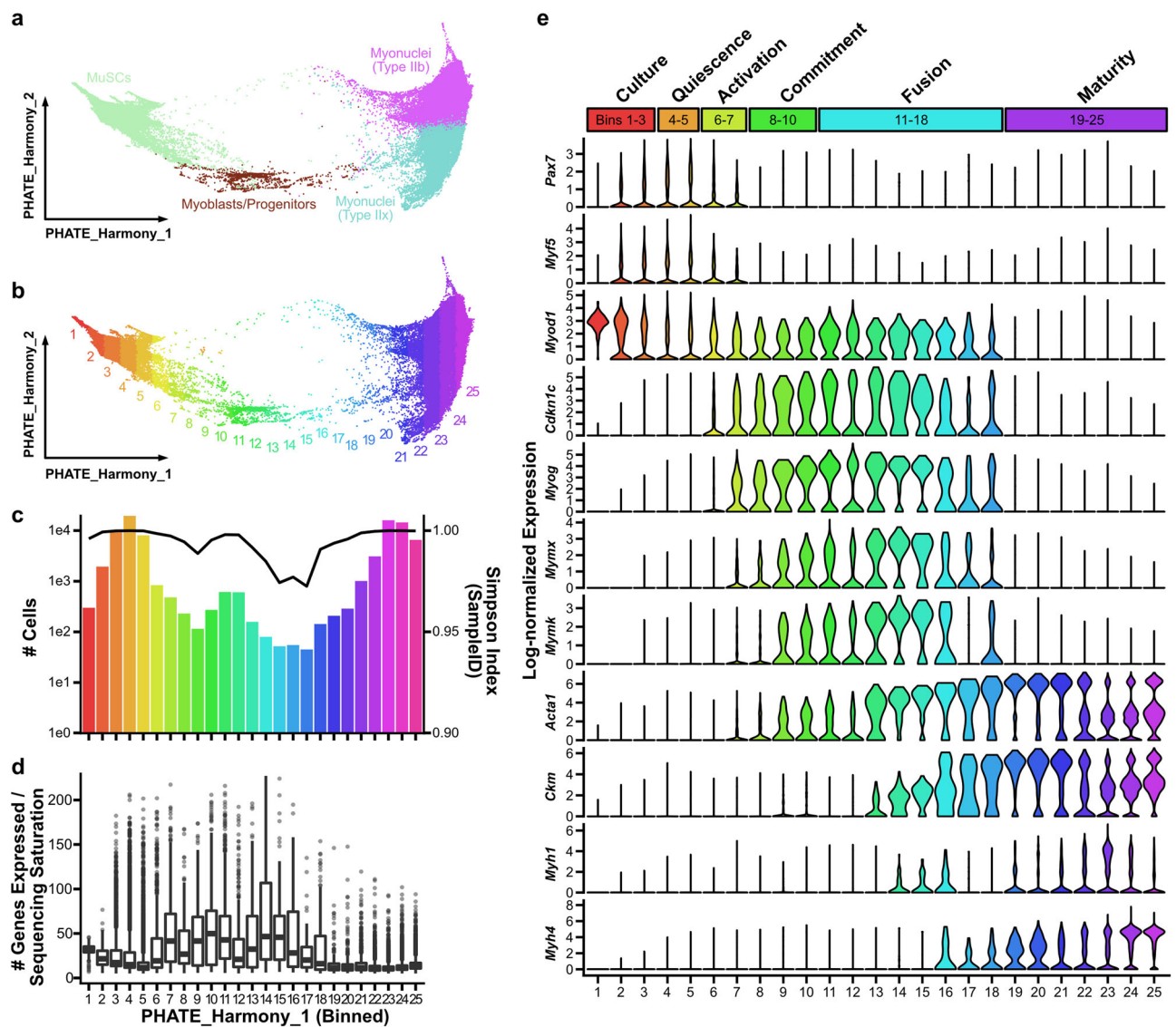

**Fig. 2 Construction of a densely sampled model of myogenesis reveals transcriptomic heterogeneity in intermediate cell states. a** 84,383 myogenic and myofiber cells were selected from the integrated compendium and dimensionally reduced with PHATE to produce a consensus differentiation trajectory. **b** Cells were binned along the first PHATE axis (PHATE_Harmony_1). **c** Cell counts within each bin are shown (bar plot). Simpson's diversity coefficients were computed on sample identifiers to determine data-source complexity for each bin (line plot). **d** Transcriptomic diversity by bin, reported as the number of genes detected per cell normalized to the sample sequencing saturation to account for differences in sequencing depth. Boxplot shows the median and quartile values. **e** Log-normalized expression levels of canonical myogenic genes, reported as violin plots for each PHATE bin, with myogenic state listed at top.

**Deconvolution of spatial RNA sequencing data using a large-scale, integrated reference.** scRNAseq methods do not recover spatial information. Current methods for spatial RNA sequencing on the other hand are limited either by the number of features they can detect or the spatial resolution they can achieve[46]. Here, we performed spatial RNA sequencing (Visium, 10x Genomics), and used the sc/snRNAseq dataset to deconvolute low-resolution spatial RNA sequencing data and identify which cell types localize together during muscle injury response. To enrich the intermediate myogenic cell populations, we performed spatial RNA-seq on samples collected at two, five, and seven days after chemical injury with notexin (Fig. 4a). Within the muscle injury zone, we observed transcripts associated with MuSC activation (*Myod1*) and a loss of mature myosin expression (*Myh1*) at 2 dpi followed by expression of cell cycle inhibitors (*Cdkn1c*), myogenic commitment markers (*Myog*), and fusogens (*Mymk*, *Mymx*) at 5 dpi (Fig. 4b, c).

Each Visium spot is 55 μm in diameter and therefore averages the expression profiles from multiple cells. To deconvolute the expression profile of each spot we used BayesPrism, a Bayesian algorithm designed to estimate cell type composition within a bulk RNAseq dataset using a single-cell reference as prior information[47]. We treated each individual spot as a bulk RNAseq dataset and used BayesPrism to estimate what fractions of the transcripts (theta values) within each spot are derived from the different cell types represented in the scRNAseq reference. We defined the myogenic cell types by incorporating PHATE binning labels (Fig. 2b) into the single-cell reference compendium (Fig. 1d). We relabeled MuSCs and Myoblasts/Progenitors (Fig. 1d) as Quiescent MuSCs, Activated MuSCs, Committed Myoblasts, or Fusing Myocytes, according to the PHATE bin they occupied. This cell-type deconvolution confirmed the enriched abundance of fusing myocytes within the injury zone at 5–7 dpi with robust spatial gradient not captured by any individual

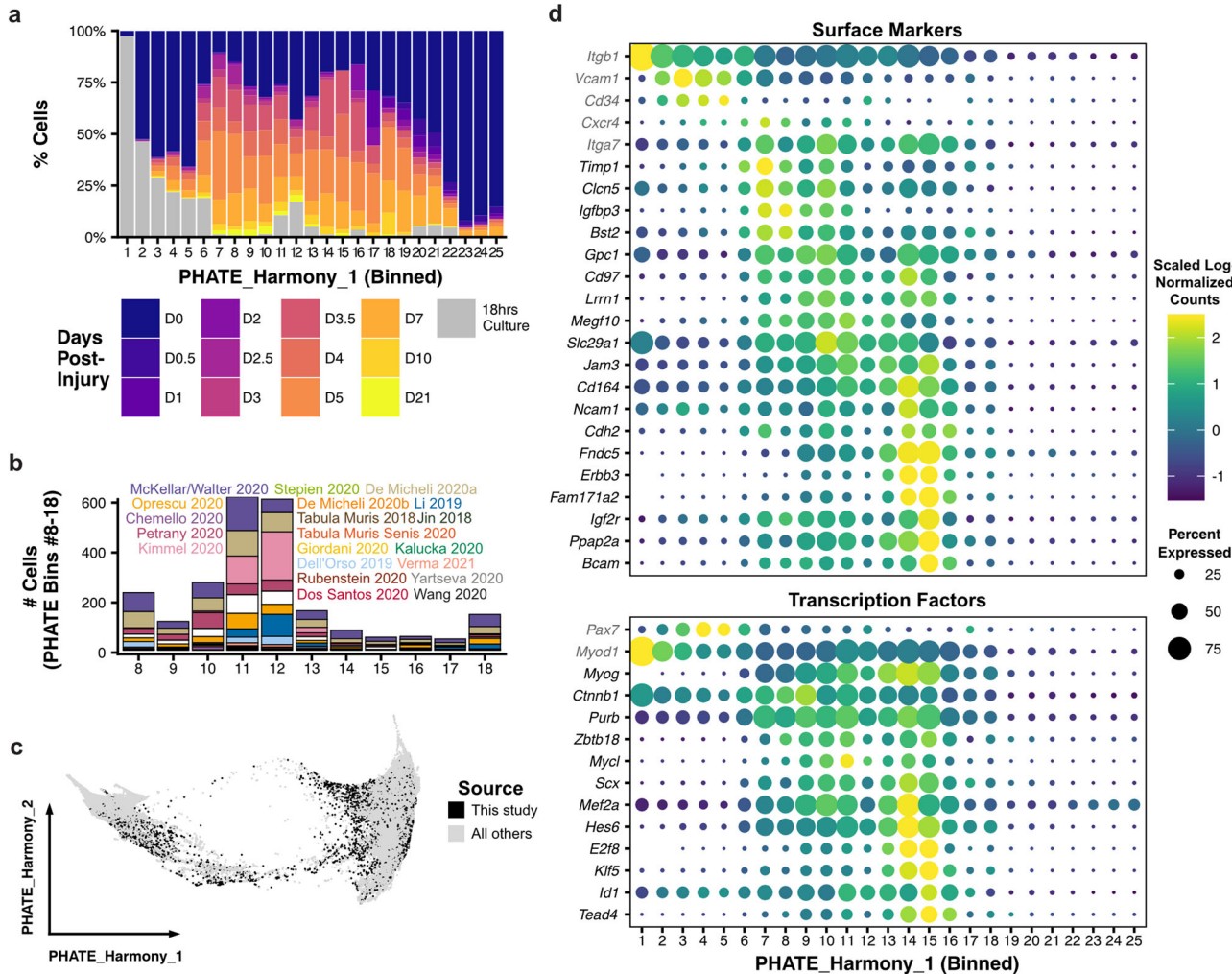

**Fig. 3 Identification of sparsely transcribed genes marking transient states in myogenesis. a** Cellular composition of each myogenic PHATE bin (see Fig. 2b, c) colored by the sample's injury time-point. **b** The subset of committed and fusing cells (bins 8–18; see Fig. 2e) were examined. The number of cells in each PHATE bin is plotted as a stacked bar, colored by dataset source (see Fig. 1c). **c** PHATE embedding of the myogenic cells, with the 7,221 myogenic cells newly generated in this study. **d** Dot plots showing the expression frequency and magnitude of surface markers (*top*) and transcription factors (*bottom*) genes in each PHATE bin. Average gene expression values are scaled (via Seurat). Canonical (*Itgb1, Vcam1, Cd34, Cxcr4, Itga7, Pax7, Myod1, Myog*; gene names in gray) and select differentially expressed genes (gene names in black) are reported.

commitment gene (Fig. 4e). Likewise, we observed an increase within the injury site of pro-remodeling FAPs (*Pdgfra*hi/*Cd34*lo/*Mmp14*hi/*Col3a1*hi) during injury response and a decrease in patrolling monocytes (*Fabp5*hi/*Cx3cr1*neg/*Ly6c2*neg) over the injury time-course (Fig. 4b, e).

Spatial RNAseq analyses can inform the spatial localization of cell types, and their co-occurrence with other cell types during injury response independent of dissociation-related artefacts (Fig. 4d, e). Based on the BayesPrism theta fractions, we enumerated the co-occurrence of cell types within each spot (Fig. 4f). We asked if the relative abundance and localization of myogenic cell types inferred by BayesPrism reflects a pattern of coordinated myogenic commitment, and how it changes with time post-injury (Fig. 4g). As expected, we found that myogenic cells frequently co-localize with mature myofibers across all timepoints. We observed that the fusing myocyte subset decreases in co-occurrence with patrolling monocytes over the time course, but increases co-occurrence with both M2 macrophage subsets, matching prior understanding of the dynamics and commitment associations of these immune cell populations[48,49].

We next asked whether the single-cell reference compendium could be used to identify potential interactions between the cell types which are spatially co-occurrent with myogenic cells during injury response. We applied CellChat[50], a tool for inference, statistical analysis, and visualization of cell-cell interactions, to prominent cell type ensembles within the integrated sc/snRNAseq data. First, we found that the three FAP subtypes and tenocytes had the highest predicted interaction strengths with myogenic cells, suggesting they exhibit the highest co-transcription of ligand-receptor pairs (Fig. 5a). We then focused our analysis on FAPs because of their patterns of co-occurrence with myogenic cells (Fig. 4f, g). We found that most of the predicted interactions between FAPs and myogenic cells were classified as "Secreted Signaling" (670/1153; 58%) or "ECM-Receptor" (419/1153; 36%), while "Cell-Cell Contact" represented a low proportion of predicted interactions (64/1153; 5.6%).

Due to its highest cumulative interaction strength in the CellChat analysis, we further explored Midkine (labeled as "MK" in the CellChat database) signaling interactions between pro-remodeling FAPs and myogenic cells (Fig. 5b). We found that

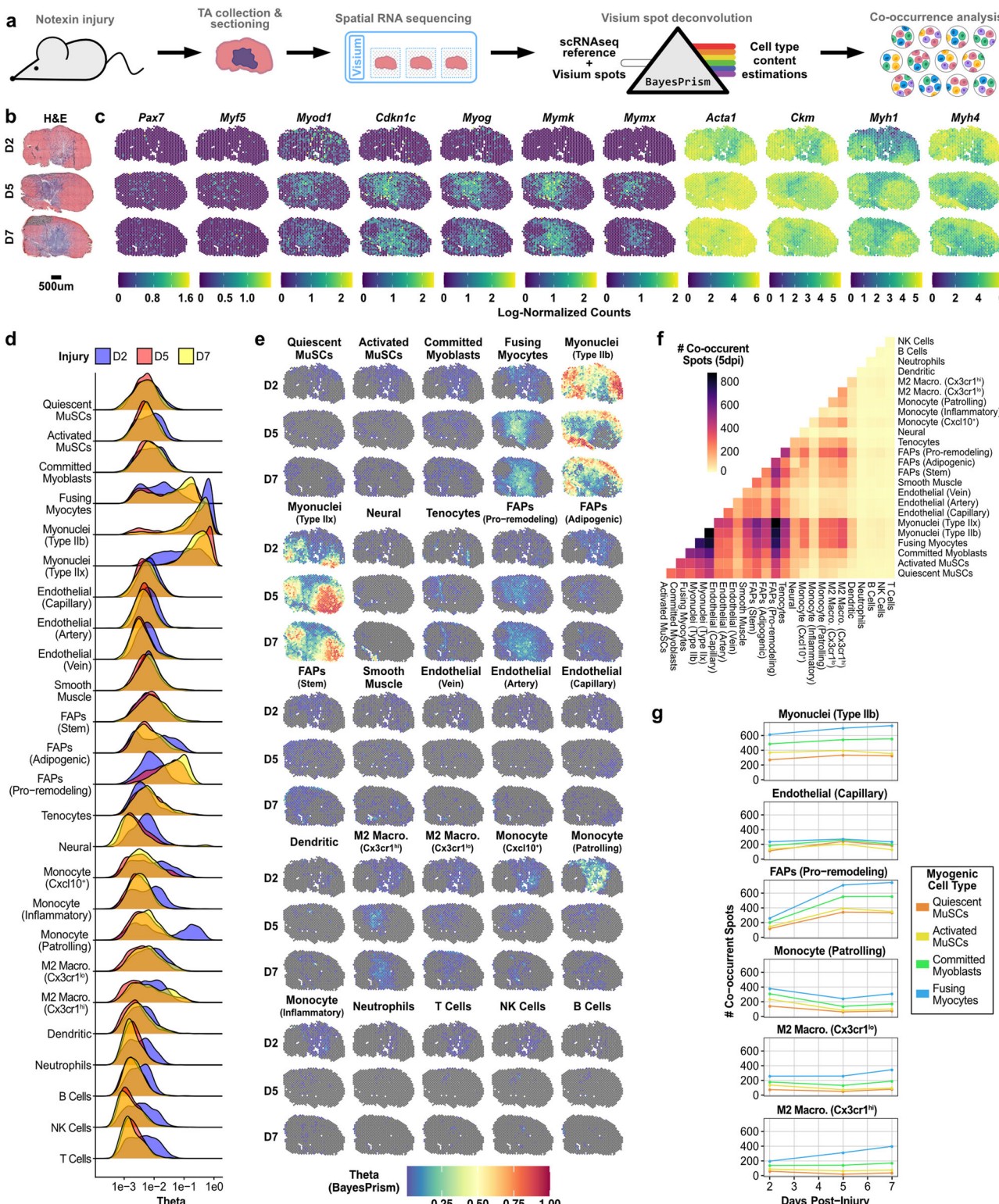

**Fig. 4 Deeply profiled cell types enable deconvolution of spatial RNA-sequencing data. a** Workflow for the generation of Visium spatial RNA sequencing of regenerating skeletal muscle and spot deconvolution to cell subtype annotations from the sc/snRNA-seq compendium. **b** H&E images of mouse tibialis anterior samples at two, five, and seven days post-notexin injury used for spatial RNA sequencing. **c** Spatial expression patterns of canonical myogenesis gene expression by Visium spot. The color scale shows the log-normalized counts for each spot. **d** Histogram of inferred cell type abundance, via BayesPrism, in each spot, colored by injury response time-point. Theta values reflect the estimated fraction of transcripts attributed to each cell type. **e** Inferred cell type content in each Visium spot. **f** Spot co-occurrence between each of the cell types. Injury time-points reported separately. **g** Spot co-occurrence of select cell types with myogenic cell subtypes by injury time-point.

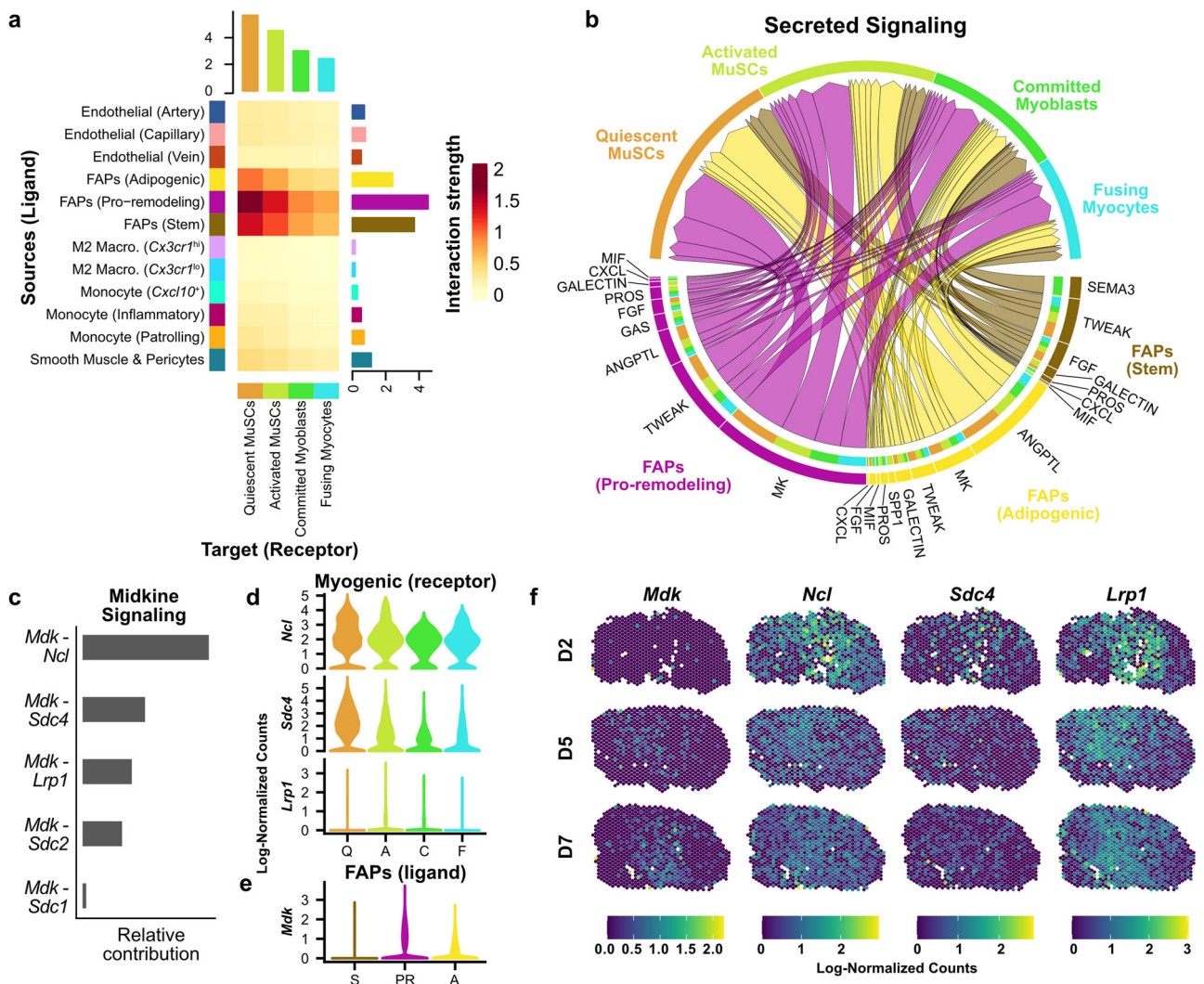

**Fig. 5 Integrated ligand-receptor analysis in sc/snRNAseq data compendium informs colocalized communication analysis in spatial transcriptomics data. a** Predicted ensemble ligand-receptor interactions strengths between endothelial, FAP, myeloid, smooth muscle (ligand-expressing), and myogenic (receptor-expressing) cell types using CellChat. **b** Chord plot showing the significant pathways annotated as "Secreted Signaling" between FAP and myogenic sub-populations, as assessed by CellChat (see "Methods"). Chords point from the FAP ligand source (protein name abbreviations listed; MK, midkine) to the myogenic cell sub-population expressing its partner receptor (names not listed). The width of the chords shows the interaction strength. **c** Relative interaction strengths of specific Midkine ligand-receptor pairs summed across all FAP and myogenic populations. **d** Log-normalized expression of the receptor and **e** ligand gene transcripts are shown for the sc/snRNAseq data compendium, for the four myogenic cell types, and the three FAP cell types, respectively. **f** Log-normalized expression of Midkine signaling factors in spatial RNAseq data (at 2, 5, and 7 dpi).

*Mdk* (the transcript encoding the Midkine protein) was highly expressed in pro-remodeling FAPs and that Midkine receptor genes (*Ncl*, *Sdc4*, and *Lrp1*) were highly expressed in multiple myogenic cell types, especially quiescent MuSCs (Fig. 5c–e). We then examined *Mdk* expression patterns in the spatial RNAseq data and observed that *Mdk* expression increased during injury response, in a manner spatially coincident with the expression of the *Ncl* and *Lrp1* Midkine receptor genes (Fig. 5f). Overall, these analyses demonstrate the utility of a large-scale single-cell reference transcriptome, together with spatial RNAseq analyses, to map the spatiotemporal coordination of infrequent cell types and states involved in skeletal muscle repair.

## Discussion

scRNAseq enables a high-resolution view of cellular dynamics in muscle injury response. However, current scRNAseq methods are ill-suited to study rare and transient cell populations due to limitations on the number of cells captured from each sample[51]. Here, we demonstrate that large-scale integration of newly generated and public single-cell transcriptomic data can overcome this limitation. We provide a scalable solution to unify sc/snRNAseq data from diverse experimental settings and techniques. We annotated cell subtypes using a curated panel of 112 marker genes from the literature[11,15,16,20,41,49,52–61]. We observed distinct subpopulations of FAPs, endothelial cells, and immune cells which were not consistently represented in prior sc/snRNAseq studies[15–17]. For example, immune cells distinctly split into neutrophils, dendritic cells, three populations of monocytes (patrolling, inflammatory, *Cxcl10*[hi]), two populations of M2 (anti-inflammatory) macrophages, and separate clusters of B, T, and NK cells, which are the primary immune cell constituents in muscle repair[48,62].

Large-scale integration of scRNAseq data generated a continuous, transcriptomic model of myogenesis, including rare

transitional myogenic cells. We identified candidate surface markers and transcription factor regulators distinct to the stages of myogenic commitment and myocyte fusion (represented by PHATE bins 8–10 and 11–18, respectively), which could not be resolved with individual datasets. Interestingly, we found signatures of surface receptors, such as *Erbb3* and *Cd97*, specific to fusing myocytes (bins 11–18), which may enable improved prospective isolation strategies compared to less stage-specific cell markers like β1-integrin (*Itgb1*). Notably, ERBB3/HER3 (encoded by *Errb3*) has been identified as a myogenic progenitor marker of human pluripotent stem cell-derived myogenic progenitors[63]. Similarly, we observed a set of transcription factors with specific expression at the commitment/fusion stages, including *E2f8*, *Tead4*, and *Mef2a*. MEF2A has been reported to be a myogenic commitment regulator[23,64]. Intriguingly, Tead4 is required for myoblast differentiation and binds the *E2f8* gene, suggesting that E2F8 may help mediate TEAD4-promoted myogenic commitment[65,66]. More generally, we found that myogenesis is characterized by a wave of transcriptomic diversification. We observed a larger number of RNA transcripts per cell at the progenitor commitment and fusion stages than in muscle stem cells or myofibers. This agrees with the observation that stem/progenitor cell trajectories exhibit changes in transcriptional diversity across their maturation axis, but differs with specific trends found in hematopoesis[43]. Last, we generated the first, to the best of our knowledge, transcriptome-level spatial RNAseq dataset of regenerating murine skeletal muscle. We repurposed BayesPrism[47], a recently developed algorithm for the deconvolution of bulk RNAseq datasets, to estimate the cell composition of each spot and identified putative cellular interactions within those spots that may drive myogenesis. We further explored these interactions by applying CellChat[50] to identify active ligand-receptor interactions between spatially co-occurrent cell types. We found that FAPs had the most predicted interactions with myogenic cells and that those interactions diminish for more differentiated myogenic stages. We showed that paired scRNAseq and spatial RNAseq provide synergies for studying cell−cell interactions by exploring Midkine signaling, the highest-ranked ligand from the FAP-myogenic interactions in the CellChat analysis and with receptors highly expressed (*Ncl* and *Lrp1*) within multiple myogenic sub-populations. Midkine signaling has been implicated in limb regeneration[67], neuromuscular[68] and epicardial development[69], and stem cell proliferation[68,70,71]. Using our spatial RNAseq data, we examined spatiotemporal patterns of Midkine-related gene expression in skeletal muscle regeneration and found that *Mdk* is spatially co-expressed with *Ncl* and *Lrp1*, suggesting the possibility of a locally coordinated paracrine signaling system.

The past decade has shown rapid growth in the number of cells that can be assayed in each experiment. Recent methods which utilize combinatorial indexing can yield more than $10^6$ transcriptomes per experiment[72,73]. Unfortunately, these methods are difficult to implement, produce a relatively low depth of coverage for each cell, and require huge amounts of sequencing. These factors lead to large costs which are often outside the budget of individual labs. Large collaborative efforts such as the Tabula Muris Senis or Human Biomolecular Atlas Program have generated massive reference transcriptome datasets, but have done so only for a limited number of tissues and disease settings[31,32,74]. We propose large-scale integration of publicly available data as the most economical and effective method for generating consortium-level reference transcriptomes. Large-scale integration enables the incorporation of sample diversity into reference transcriptomes, which will likely better reflect the underlying biology across individuals. Taken together, our study supports the utility of large-scale integration of single-cell transcriptomic data as a powerful tool for biological discovery. We created a public web tool (scmuscle.bme.cornell.edu/) to enable access and further interrogation of the dataset reported here.

## Methods

**Mice**. The Cornell University Institutional Animal Care and Use Committee (IACUC) approved all animal protocols, and experiments were performed in compliance with its institutional guidelines. Adult C57BL/6J mice (*mus musculus*) were obtained from Jackson Laboratories (#000664; Bar Harbor, ME) and were used at 4–7 months of age. Aged C57BL/6J mice were obtained from the National Institute of Aging (NIA) Rodent Aging Colony and were used at 20 months of age. For new scRNAseq experiments, female mice were used in each experiment.

**Mouse injuries and single-cell isolation**. To induce muscle injury, both tibialis anterior (TA) muscles of old (20 months) C57BL/6J mice were injected with 10 µl of notexin (10 µg/ml; Latoxan; France). At 0, 1, 2, 3.5, 5, or 7 days post-injury (dpi), mice were sacrificed and TA muscles were collected and processed independently to generate single-cell suspensions. Muscles were digested with 8 mg/ml Collagenase D (Roche; Switzerland) and 10 U/ml Dispase II (Roche; Switzerland), followed by manual dissociation to generate cell suspensions. Cell suspensions were sequentially filtered through 100 and 40 µm filters (Corning Cellgro #431752 and #431750) to remove debris. Erythrocytes were removed through incubation in erythrocyte lysis buffer (IBI Scientific #89135-030).

**Single-cell RNA-sequencing library preparation**. After digestion, single-cell suspensions were washed and resuspended in 0.04% BSA in PBS at a concentration of $10^6$ cells/ml. Cells were counted manually with a hemocytometer to determine their concentration. Single-cell RNA-sequencing libraries were prepared using the Chromium Single Cell 3′ reagent kit v3 (10x Genomics, PN-1000075; Pleasanton, CA) following the manufacturer's protocol. Cells were diluted into the Chromium Single Cell A Chip to yield a recovery of 6,000 single-cell transcriptomes. After preparation, libraries were sequenced using a NextSeq 500 (Illumina; San Diego, CA) using 75 cycle high output kits (Index 1 = 8, Read 1 = 26, and Read 2 = 58). Details on estimated sequencing saturation and the number of reads per sample are shown in Supplementary Data 1.

**Spatial RNA sequencing library preparation**. Tibialis anterior muscles of adult (5 mo) C57BL6/J mice were injected with 10 µl notexin (10 µg/ml) at 2, 5, and 7 days prior to collection. Upon collection, tibialis anterior muscles were isolated, embedded in OCT, and frozen fresh in liquid nitrogen. Spatially tagged cDNA libraries were built using the Visium Spatial Gene Expression 3′ Library Construction v1 Kit (10x Genomics, PN-1000187; Pleasanton, CA) (Fig. S7). Optimal tissue permeabilization time for 10 µm thick sections was found to be 15 min using the 10x Genomics Visium Tissue Optimization Kit (PN-1000193). H&E stained tissue sections were imaged using Zeiss PALM MicroBeam laser capture micro-dissection system and the images were stitched and processed using Fiji ImageJ software. cDNA libraries were sequenced on an Illumina NextSeq 500 using 150 cycle high output kits (Read 1 = 28 bp, Read 2 = 120 bp, Index 1 = 10 bp, and Index 2 = 10 bp). Frames around the capture area on the Visium slide were aligned manually and spots covering the tissue were selected using Loop Browser v4.0.0 software (10x Genomics). Sequencing data were then aligned to the mouse reference genome (mm10) using the spaceranger v1.0.0 pipeline to generate a feature-by-spot-barcode expression matrix (10x Genomics).

**Download and alignment of single-cell RNA sequencing data**. For all samples available via SRA, parallel-fastq-dump (github.com/rvalieris/parallel-fastq-dump) was used to download raw.fastq files. Samples that were only available as.bam files were converted to.fastq format using bamtofastq from 10x Genomics (github.com/10XGenomics/bamtofastq). Raw reads were aligned to the mm10 reference using cellranger (v3.1.0).

**Preprocessing and batch correction of single-cell RNA sequencing datasets**. First, ambient RNA signal was removed using the default SoupX (v1.4.5) workflow (autoEstCounts and adjustCounts; github.com/constantAmateur/SoupX). Samples were then preprocessed using the standard Seurat (v3.2.1) workflow (NormalizeData, ScaleData, FindVariableFeatures, RunPCA, FindNeighbors, FindClusters, and RunUMAP; github.com/satijalab/seurat). Cells with fewer than 750 features, fewer than 1000 transcripts, or more than 30% of unique transcripts derived from mitochondrial genes were removed. After preprocessing, DoubletFinder (v2.0) was used to identify putative doublets in each dataset, individually. $BC_{mvn}$ optimization was used for $P_K$ parameterization. Estimated doublet rates were computed by fitting the total number of cells after quality filtering to a linear regression of the expected doublet rates published in the 10x Chromium handbook. Estimated homotypic doublet rates were also accounted for using the modelHomotypic function. The default $P_N$ value (0.25) was used. Putative doublets were removed from each individual dataset. After preprocessing and quality filtering, we merged the datasets and performed batch-correction with three tools, independently—Harmony (github.com/immunogenomics/harmony) (v1.0), Scanorama (github.com/brianhie/scanorama) (v1.3), and BBKNN (github.com/Teichlab/bbknn) (v1.3.12). We then used Seurat to process the integrated data. After initial integration, we removed the noisy cluster and re-integrated the data using each of the three batch-correction tools.

**Cell type annotation**. Cell types were determined for each integration method independently. For Harmony and Scanorama, dimensions accounting for 95% of the total variance were used to generate SNN graphs (Seurat::FindNeighbors). Louvain clustering was then performed on the output graphs (including the corrected graph output by BBKNN) using Seurat::FindClusters. A clustering resolution of 1.2 was used for Harmony (25 initial clusters), BBKNN (28 initial clusters), and Scanorama (38 initial clusters). Cell types were determined based on the expression of canonical genes (Fig. S3). Clusters that had similar canonical marker gene expression patterns were merged.

**Pseudotime workflow**. Cells were subset based on the consensus cell types between all three integration methods. Harmony embedding values from the dimensions accounting for 95% of the total variance were used for further dimensional reduction with PHATE, using phateR (v1.0.4) (github.com/KrishnaswamyLab/phateR).

**Deconvolution of spatial RNA sequencing spots**. Spot deconvolution was performed using the deconvolution module in BayesPrism (previously known as "Tumor microEnvironment Deconvolution", TED, v1.0; github.com/Danko-Lab/TED). First, myogenic cells were re-labeled, according to binning along the first PHATE dimension, as "Quiescent MuSCs" (bins 4–5), "Activated MuSCs" (bins 6–7), "Committed Myoblasts" (bins 8–10), and "Fusing Myocytes" (bins 11–18). Culture-associated muscle stem cells were ignored and myonuclei labels were retained as "Myonuclei (Type IIb)" and "Myonuclei (Type IIx)". Next, highly and differentially expressed genes across the 25 groups of cells were identified with differential gene expression analysis using Seurat (FindAllMarkers, using Wilcoxon Rank Sum Test; results in Supplementary Data 2). The resulting genes were filtered based on average $\log_2$-fold change (avg_logFC > 1) and the percentage of cells within the cluster which express each gene (pct.expressed > 0.5), yielding 1,069 genes. Mitochondrial and ribosomal protein genes were also removed from this list, in line with recommendations in the BayesPrism vignette. For each of the cell types, mean raw counts were calculated across the 1,069 genes to generate a gene expression profile for BayesPrism. Raw counts for each spot were then passed to the run.Ted function, using the "GEP" option for input.type and default parameters for the remaining inputs. Final Gibbs theta values were used as estimates for the fraction of transcripts from each spot that were derived from each of the 25 cell types.

**Spot co-occurrence of cell subtypes**. Cell type co-occurrence was computed as before by Mantri et al.[69]. Briefly, for each spot cell types were ordered according to the computed theta values (estimated percentage of reads attributed to a cell type). Up to 10 cell types with the highest theta values were tallied for each spot. Cell types with theta values lower than 0.01 (1% of transcripts) were not counted.

**Ligand-receptor analysis with CellChat**. Ligand-receptor analysis and visualization were performed using CellChat v1.1.0 (github.com/sqjin/CellChat). The cell type labels used were derived from the Harmony integration results using all single-cell and single-nucleus data sources. Myogenic cell types were derived from binning along the PHATE embedding (Fig. 2b). Default values were used for the parameterization of each step.

**Statistics and reproducibility**. For the newly generated scRNAseq data, we collected three or four replicates for each injury time point, for a total of 23 samples. Each replicate was an independent experiment, where animals were injured, sacrificed, and analyzed individually. 88 public datasets were also downloaded, for a total of 111 scRNAseq and snRNAseq samples analyzed in this study. For spatial RNAseq, one sample was collected for each injury timepoint. The details for statistical analyses performed in this study are provided in the respective sections of the "Results" and "Methods". For differential gene expression, a Wilcoxon Rank Sum test was used (via Seurat[36]) to assess the statistical significance of gene expression differences across cell clusters.

**Reporting summary**. Further information on research design is available in the Nature Research Reporting Summary linked to this article.

## Data availability

All source data are deposited and available in public repositories. A complete list of metadata and GEO and ArrayExpress accession information for the publicly available scRNAseq and snRNAseq data[9,11,15–17,20–32] can be found in Table S1. Single-nucleus RNA sequencing data were kindly provided by the Millay lab[11], prior to public release. Newly collected scRNAseq data for two samples from 7 mo mice are deposited in GEO under accession GSE159500. The scRNAseq data from 20 mo mice are deposited in GEO under accession GSE162172. These spatial RNA sequencing data are deposited under GSE161318. SRR numbers for downloading each individual sample are compiled in the Sup. Data 1 file. Fully processed Seurat and CellChat objects for the entire compendium (Fig. 1), myogenic cells (Figs. 2, 3), spatial RNAseq datasets (Fig. 4), and ligand-receptor analysis (Fig. 5) are available for download on Dryad (doi:10.5061/dryad.t4b8gtj34).

Results from differential expression analyses associated with these figures are compiled in the Supplementary Data 2−6 files.

## Code availability

All code for processing and analysis of the scRNAseq and spatial RNA sequencing data, as well as supplementary data and gene lists used in this study, are available on Github (github.com/mckellardw/scMuscle) with the identifier (https://zenodo.org/badge/latestdoi/309225505). The full integrated dataset with visualization tools is available at scmuscle.bme.cornell.edu.

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

## Acknowledgements
We thank Peter Schweitzer and colleagues in the Cornell Biotechnology Resource Center for their help with preparing the Chromium and Visium datasets. We thank the Cornell Center for Animal Resources and Education for animal housing and care. We thank Cornell Red Cloud Computing Services for website support. We thank Andrea De Micheli, Ern Hwei Hannah Fong, and Alexandra Dalaya for helping with mouse procedures and generating single-cell RNA sequencing datasets. We thank all investigators who made their datasets public, especially Douglas Millay for providing the single-nucleus RNA sequencing data before its publication and helpful discussions. We thank Alexandre P. Cheng, Benjamin Grodner, Hao Shi, Charles Heinke, Emily Laurilliard, Umji Lee, Tinyi Chu, and Charles Danko for helpful discussions and feedback. This work was supported by the US National Institutes of Health (NIH) grant 1DP2AI138242 to I.D.V., and NIH grant R01AG058630 to B.D.C and I.D.V., and T32EB023860 to D.W.M. The content is solely the responsibility of the authors and does not necessarily represent the official views of the NIH.

## Author contributions
D.W.M., I.D.V., and B.D.C. designed the study. D.W.M. and L.D.W. carried out the experiments. D.W.M., M.M., M.F.Z.W., and L.D.W. analyzed the data. L.T.S. and D.W.M. designed the web resource. D.W.M., I.D.V., and B.D.C. wrote the manuscript. All authors provided feedback and comments.

## Competing interests
The authors declare no competing interests.
