## [Transparent Peer Review File · Communications Biology]

Reviewers' comments:

Reviewer #1 (Remarks to the Author):

The authors took advantage of the recent publications of various single-cell and single-nuclei RNA-seq data from muscle in order to get a comprehensive view of the transcriptional states of cells present within the muscle. This study presents a valuable resource to the scientific community by aggregating myriad datasets to help scientists in the field explore various transcriptional states of number cell populations in injured and regenerating muscle. Furthermore, the authors highlight novel findings in muscle stem cells that were not previously identifiable due to small cell numbers, which has been overcome with aggregating the data provided. Including some experimental data to understand the spatial organization of cells participating in muscle regeneration further supports the combination of high-throughput sequencing and analysis techniques to help us understand the players involved in successful muscle regeneration. Additionally, the interactive tool provided by the authors should be a valuable resource to the scientific community and makes these findings accessible to a broad audience.

Although the dearth of single-cell datasets is ever expanding, the importance and information garnered from these datasets can at times be difficult to outline. While the authors perform substantial data analysis to try and outline some information from all the datasets published, the overarching conclusions that can be made from these analyses are not clearly established. Indeed, the authors identify potential surface markers that can be used to potentially isolate muscle stem cells that are committed to the myogenic program, however more analysis could be performed to gain more insight into aspects regulating regeneration.

Comments/revisions

1. Can the authors include more data analysis on the FAPs and immune cell subsets? For example, what aspects of FAPs may represent a true 'progenitor' population, are there specific markers for these and is this population homogenous? For immune cell markers, are there any signatures that can be used to identify specific subsets, or markers that can be used to identify and track tissue resident macrophages? Perhaps a section that outlines a handful of these aspects would be helpful for the readers to understand the broad impact of the study and potential usefulness of the resource outlined.
2. The authors identify number cell types that are co-occurrent during the regeneration process. Are there any receptor-ligand pairs that may be enriched between these cells that can be linked to some of the cellular processes involved in muscle regeneration? For example, are there any ligands from cells that co-occur with macrophages that may trigger a pro to anti-inflammatory phenotype change?

Reviewer #2 (Remarks to the Author):

This manuscript by McKellar et al. described the methods and findings to integrate datasets of single cell- or single nuclear-based transcriptome analyses of skeletal muscle in homeostatic conditions and following injury, from both published studies and their own study. By doing this, the scale of the analysis was expanded greatly that these data enabled identification of the predominant cell types in skeletal muscle, and resolved multiple cell subtypes. The representation of different experimental conditions and the depth of transcriptome coverage enabled robust profiling of sparsely expressed genes. With this integrated analysis, the authors built a densely sampled transcriptomic model of myogenesis and identified rare, transitional states of progenitor commitment and fusion that are poorly represented in individual datasets. Using the integrated dataset as a reference, the authors also achieved a high-resolution, local deconvolution of cell subtypes in myogenic differentiation by spatial RNA sequencing. These findings supported the utility of large-scale integration of single-cell transcriptomic data as a tool for biological discovery. Overall, this is a very interesting and innovative

study, which shows the power of integration of scRNA-seq data, snRNA-seq data and Visium spatial RNA sequencing data, as well as cutting-edge technologies with novel bioinformatics tools (batch effect correction algorithms) to predict and analyze muscle repair, myogenic differentiation etc. These may lead to important insights into both bioinformatical and clinical aspects.

The most critical basis of the presented study is the proper integration of published datasets of variable sources, which are undoubtedly impacted by variable experimental conditions. For this reason, a detailed, well-reasoned description and validation of the tools used for batch effect correction must be clearly presented to the extent that it is comprehensible to ordinary readers without in-depth knowledge of bioinformatics. The current manuscript needs further improvement to reach this goal.

Major concerns:

1) scRNA-seq measures both cytoplasmic and nuclear transcripts, while snRNA-seq mainly measures nuclear transcripts. In this study, scRNA-seq and snRNA-seq datasets were integrated for the analysis. This could be problematic because the nuclear transcripts profile could be very different from the cytoplasmic one even in the same functional sub-cell types, leading to big bias for the following clustering analysis for cell types and subtypes. The authors should provide evidence that the integration of these two different types of analysis is valid and explain how the potential bias is corrected.

2) There are more than 14 batch effect correction methods reported (references 18 and 19). Only 3 methods were compared in this study however, and the authors concluded that Harmony is the best one. The authors should provide reasons why these 3 methods were chosen for comparison. The authors also need to include descriptions of the criteria used to judge that Harmony is best method comparing to the other two. In addition, reference 19 is a preprint without peer review, it may not be appropriate to cite this reference.

3) UMAPs of scRNAseq show nicely the separation of cell types in Figure1D (and figure S2) after integration with Harmony; however, a control UMAP showing results with no batch effect correction should be included. Similarly, Fig1C should also include a UMAP showing results after integration with batch effect correction.

4) Some results are not clearly described: 1) Fig3B: what do the different colors represent for, different data source? Please describe. 2): what does "cultured MuSCs" stand for, in line 163 and Figure 2E?

Minor concerns:

1) In line 197, Fig.3C should be Fig.3D according to the description.

2) Figure S1, labels D) and E) are missing in Fig1SD and Fig1SE.

3) Figure S2, labels for all clusters should be provided in all figures, as being provided in the Harmony analysis figures.

4) FigureS3, labels A), B) and C) are missing.

5) In line 296, "... We created a public web tool (scmuscle.bme.cornell.edu) to enable...", the website link is not working.

Reviewer #3 (Remarks to the Author):

In the manuscript "Strength in numbers: Large-scale integration of single-cell transcriptomic data captures transitional progenitor states in muscle regeneration", McKellar et al. very efficiently integrate single-cell (scRNAseq) and single-nuclei (snRNAseq) RNAseq data from 23 newly collected and 88 publicly available datasets. The analysis covers two important aspects: it allows to follow the transitional states of the transcriptome during myogenesis (stem cells in a quiescent status, committed progenitors, transitional fusion states to myofiber maturation), it captures the predominant cell types present in skeletal muscle and involved in muscle repair.

The authors add the first spatial RNAseq report, performed at different time points after muscle injury. There is no doubt that the pieces of information reported in the manuscript are of value for the muscle field. The comprehensive view of the muscle transcriptome at different stages, and the web tool

publicly available will help to enhance our knowledge of muscle biology. As such, the manuscript is a good candidate for publication in Communications Biology. Clarifications of some points will help to prove some of the findings.

1. The pipeline and methods used to combine different datasets and limit batch effects are well explained and convincing. In reference to Fig S1, it will benefit the readers if the authors can include a more detailed analysis of scRNAseq vs snRNAseq. Do they show a comparable induction of dissociation-induced stress genes? Are the major RNA signatures conserved?
2. The idea of identifying new surface markers to enrich for rare transient cell types is very interesting. It will be valuable to test some of the listed markers in a FACS isolation followed by bulk RNAseq to have a more comprehensive view of the transcriptome of transient committed/fusing cells. Alternatively, combined immunostaining for some of the newly identified surface markers can strengthen the conclusions of their bioinformatic analysis.
3. The large-scale integration of single-cell transcriptomic data led to the identification of a core of transcription factors highly expressed during the commitment and fusion stages of myogenesis. Can the authors identify any network of key genes or pathways controlled by the listed transcription factors?
4. In figure 4F the authors show an interesting analysis of the co-occurrence of different cell types within each spot on the spatial transcriptomic slide. Is there any difference in the transcriptome of myogenic cells located close to macrophages, compared to myogenic cells not occupying the same spot? Is there any other cross-influence in the transcriptome between different cell types?

Summary Responses to General Comments:

Topic: Justification of use of Harmony and related batch-correction inquiries.

Response #1: We now provide a more detailed explanation of how we selected Harmony for batch correction among the alternative options, based on its computational performance and robustness and classification accuracy in cell type clusters. This is provided in revised text on lines 109-121. Further explanation is provided in Response #13 below.

Topic: Discuss differences between single-cell (sc) and single-nucleus (sn) RNA-seq data and whether integration algorithms overcome technical biases between these assay types.

Response #2: We now clarify that the use of Harmony (but not other batch-correction algorithms) is able to integrate the disparate data sources such that not outlier cell clusters appear simply due to single-cell/single-nucleus assay types (see **Fig. S2** and lines 120-121). We provide a new analysis of how assay type influences RNA capture and detection across all data sources within the compendium (see new **Figs. S5** and **S6** and lines 145-157). We conclude that assay type influences the relative capture and detection of certain types of RNA species, likely related to their cellular compartmentalization, rather uniformly across most cell types. For example, lncRNAs are generally enriched in single-nucleus data and protein coding RNAs are enriched in single-cell data, regardless of cell type. This suggests that assay technology leads to some biases in differential capture, even after ambient RNA removal (by SoupX), but does not lead to widespread cell clustering failure in the final integrated dataset when Harmony is implemented. Further explanation is provided in Responses #13 and #14 below.

Topic: Discuss cell-type-specific gene expression differences

Response #3: We now provide a more extensive presentation of cell type specific markers derived from differential gene expression analyses in new **Supplementary Files 4** (for Endothelial Cell types), **5** (for FAP cell types), and **6** (for multiple Myeloid cell types). We also provide a more detailed PHATE visualization of these cell types and their sub-populations in new Figure S4. Please refer to text lines 132-134 and Responses #9 and #20 below.

Topic: Expand the analysis of spatial transcriptomics data to consider potential ligand-receptor interactions and the spatial coordination on gene expression responses.

Response #4: The suggestion of examining the spatial variation in gene expression by cell type may provide evidence of direct cell-cell communication patterns was helpful and intriguing. The BayesPrism algorithm approach provides an estimate of relative cell type abundance within each Visium spot by deconvolution but is not able to provide an accurate estimate of the unique transcriptomic profile of each deconvolved cell. Instead, we sought to address this by examining possible ligand-receptor interactions within the single-cell data compendium and our new Visium spatial transcriptomics data. To this end, we applied a ligand-receptor interaction algorithm known as CellChat to those cell type combinations found to be highly co-occurrent within the spatial transcriptomics data. These new results are provided in new Figure 5 and text lines 263-278 and 318-328. The key observation from these analyses is of highly co-occurrent expression of the ligand Midkine (*Mdk*) by FAPs and its receptors (*Ncl*, *Lrp1*) by myogenic progenitor cells during muscle regeneration, suggesting a locally active paracrine signaling system. Further explanation is provided in Response #10 below.

Topic: Examine the gene networks which are potentially regulate by the candidate core transcription factors.

Response #5: We appreciate the Referee's suggestion to examine gene regulatory networks involved in the transcription factors identified by their expression patterns within the myogenic trajectory. We concluded that this would require extensive analysis and expand significantly beyond the scope of this manuscript, and therefore prioritized other revisions.

Topic: Examine and validate a subset of new surface markers suggested by this work.

Response #6: We appreciate the Referee's suggestion to validate a subset of suggested surface markers, but argue that this effort would require substantial new experiments and is beyond the scope of this revision.

Topic: Highlight all changes in the manuscript text file.

Response #7: All revisions within the manuscript file are presented in blue text for easy identification.

===

Point-by-point responses to specific Reviewer comments:

Reviewer #1 (Remarks to the Author):

The authors took advantage of the recent publications of various single-cell and single-nuclei RNA-seq data from muscle in order to get a comprehensive view of the transcriptional states of cells present within the muscle. This study presents a valuable resource to the scientific community by aggregating myriad datasets to help scientists in the field explore various transcriptional states of number cell populations in injured and regenerating muscle. Furthermore, the authors highlight novel findings in muscle stem cells that were not previously identifiable due to small cell numbers, which has been overcome with aggregating the data provided. Including some experimental data to understand the spatial organization of cells participating in muscle regeneration further supports the combination of high-throughput sequencing and analysis techniques to help us understand the players involved in successful muscle regeneration. Additionally, the interactive tool provided by the authors should be valuable resource to the scientific community and makes these findings accessible to a broad audience.

Response #8: We thank the Reviewer #1 for the appreciation of our work.

Although the dearth of single-cell datasets is ever expanding, the importance and information garnered from these datasets can at times be difficult to outline. While the authors perform substantial data analysis to try and outline some information from all the datasets published, the overarching conclusions that can be made from these analyses are not clearly established. Indeed, the authors identify potential surface markers that can be used to potentially isolate muscle stem cells that are committed to the myogenic program, however more analysis could be performed to gain more insight into aspects regulating regeneration.

Comments/revisions

1. Can the authors include more data analysis on the FAPs and immune cell subsets? For example, what aspects of FAPs may represent a true 'progenitor' population, are there specific

markers for these and is this population homogenous? For immune cell markers, are there any signatures that can be used to identify specific subsets, or markers that can be used to identify and track tissue resident macrophages? Perhaps a section that outlines a handful of these aspects would be helpful for the readers to understand the broad impact of the study and potential usefulness of the resource outlined.

Response #9: We thank Reviewer #1 for the suggestion. We agree that additional detail on other cell types would be a useful addition to this work. To achieve this, we further analyzed FAPs, myeloid immune cells (monocytes, macrophages, and dendritic cells), and endothelial cells. This analysis is shown in a new supplemental figure (**Fig. S4**). We first used PHATE to dimensionally reduce and visualize these subsets of cells (**Fig. S4A-B**). We then performed differential gene expression within each of these cell subpopulations to find specific markers for each subtype. In **Fig. S4C** we show the top 45 marker genes from this analysis. We also included the entirety of the differential gene expression results as new supplemental data files (**Sup. Files 4-6**). This analysis has deepened the utility of our work as a resource of candidate genes for others to examine subtypes of non-myogenic cells in skeletal muscle.

2. The authors identify number cell types that are co-occurrent during the regeneration process. Are there any receptor-ligand pairs that may be enriched between these cells that can be linked to some of the cellular processes involved in muscle regeneration? For example, are there any ligands from cells that co-occur with macrophages that may trigger a pro to anti-inflammatory phenotype change?

Response #10: We thank Reviewer #1 for this thoughtful suggestion. To address potential cell-cell communication that may contribute to muscle regeneration we include a new analysis which explores ligand-receptor and other interactions between cells. This work resulted in a new figure (**Fig. 5**) and text lines 263-278 and 318-328. We used CellChat (Jin et al, Nat Commun 2021) to identify communication pathways which are enhanced in specific cell type pairs within the full single-cell/single-nucleus RNA-seq compendium. CellChat identifies secreted ligand-receptor, ECM-receptor, and cell-cell contact interactions. We then explored interactions between FAP cells and myogenic cells due to the observation they had the strongest interaction strengths for secreted factors and because we previously found them to be highly co-occurrent in the spatial RNA-seq data (**Fig. 4G**). Our new analysis with CellChat suggests that they may interact with myogenic cells via secreted signaling and extracellular matrix factors, rather than cell-cell contact interactions. We then examined specific secreted ligand-receptor interactions by myogenic and FAP sub-type pairing and found an enrichment of Midkine (Mdk) related interactions. We examined the spatial RNAseq data and found that *Mdk* expression increased during injury response, in a manner spatial coincident with expression of the *Ncl* and *Lrp1* Midkine receptor genes. Overall this analysis improves our work by further demonstrating the utility of large-scale single-cell transcriptomic references for predicting pathways of cell-cell communication within complex tissues in a spatially defined manner.

Reviewer #2 (Remarks to the Author):

This manuscript by McKellar et al. described the methods and findings to integrate datasets of single cell- or single nuclear-based transcriptome analyses of skeletal muscle in homeostatic conditions and following injury, from both published studies and their own study. By doing this, the scale of the analysis was expanded greatly that these data enabled identification of the predominant cell types in skeletal muscle, and resolved multiple cell subtypes. The representation of different experimental conditions and the depth of transcriptome coverage enabled robust profiling of sparsely expressed genes. With this integrated analysis, the authors built a densely sampled transcriptomic model of myogenesis and identified rare, transitional

states of progenitor commitment and fusion that are poorly represented in individual datasets. Using the integrated dataset as a reference, the authors also achieved a high-resolution, local deconvolution of cell subtypes in myogenic differentiation by spatial RNA sequencing. These findings supported the utility of large-scale integration of single-cell transcriptomic data as a tool for biological discovery. Overall, this is a very interesting and innovative study, which shows the power of integration of scRNA-seq data, snRNA-seq data and Visium spatial RNA sequencing data, as well as cutting-edge technologies with novel bioinformatics tools (batch effect correction algorithms) to predict and analyze muscle repair, myogenic differentiation etc. These may lead to important insights into both bioinformatical and clinical aspects.

The most critical basis of the presented study is the proper integration of published datasets of variable sources, which are undoubtedly impacted by variable experimental conditions. For this reason, a detailed, well-reasoned description and validation of the tools used for batch effect correction must be clearly presented to the extent that it is comprehensible to ordinary readers without in-depth knowledge of bioinformatics. The current manuscript needs further improvement to reach this goal.

Major concerns:

1) scRNA-seq measures both cytoplasmic and nuclear transcripts, while snRNA-seq mainly measures nuclear transcripts. In this study, scRNA-seq and snRNA-seq datasets were integrated for the analysis. This could be problematic because the nuclear transcripts profile could be very different from the cytoplasmic one even in the same functional sub-cell types, leading to big bias for the following clustering analysis for cell types and subtypes. The authors should provide evidence that the integration of these two different types of analysis is valid and explain how the potential bias is corrected.

Response #11: We thank Reviewer #2 for this comment. This issue of detection biases due to single-cell assay technology is quite relevant. Indeed, we re-analyzed the data in the integrated compendium and found that there are major differences in the “biotypes” of transcripts captured in single-cell and single-nucleus data in new **Figs. 1E, S5 and S6** and text lines 145-157. We now show that both assay types capture similar levels of transcription factor transcripts across the whole compendium, which is edifying given TFs are critical for classifying cell type in clustering.

Before explaining the new results, we note that one feature of the computational workflow we use is the use of dimensional reduction strategies to reduce noise across samples. This begins by identifying the top 2000 high variance genes within the data and performing principal component analysis to distill this information into 50 dimensions. Integration is then performed in principal component space. To ensure that this strategy is not sensitive to technical differences between single-cell and single-nucleus data, we further examined the 2000 high variance genes used in our previous analyses. First, we performed differential gene expression analysis across all single-cell and single-nucleus transcriptomes. We found that only 10% of highly variable genes (199/2000) genes are differentially captured within these two data types (adjusted-p-value < 10^{-10} and log₂-fold-change > 1). Of these 199 genes, 184 of them are differentially expressed across cell types (as identified after Harmony integration; adjusted-p-value < 10^{-10} and log₂-fold-change > 1). This suggests that the differences in gene expression between cell types are greater than the differences between cells and nuclei.

To establish if there were technical biases due to the assay technology (single-cell vs. single-nucleus), we then performed differential gene expression by assay within each cell type individually (**Figs. S5 and S6**). We focused on cell types for which we have more than 200

single-nucleus transcriptomes (see **Fig. S6A**) and compared the single-nucleus transcriptomes to single-cell data (not including FACS-associated samples to exclude stress artefacts from the analysis). We conclude that assay type influences the relative capture and detection of certain types of RNA species (lncRNAs, protein coding, mitochondrial, ribosomal, heat-shock, and surface protein transcripts), likely related to their cellular compartmentalization, rather uniformly across most cell types. In contrast, transcripts encoding transcription factors are similarly detected in almost all cell types. This suggests that assay technology leads to some biases in differential capture and detection, even after ambient RNA removal (by SoupX) and batch correction) but does not lead to widespread cell clustering failure in the final integrated dataset when Harmony is implemented.

2) There are more than 14 batch effect correction methods reported (references 18 and 19). Only 3 methods were compared in this study however, and the authors concluded that Harmony is the best one. The authors should provide reasons why these 3 methods were chosen for comparison. The authors also need to include descriptions of the criteria used to judge that Harmony is best method comparing to the other two. In addition, reference 19 is a preprint without peer review, it may not be appropriate to cite this reference.

Response #13: We thank Reviewer #2 for this comment. We have updated the text in the manuscript giving clearer guidelines on how we compared the three batch-correction methods used in this study, and why we decided that Harmony was the most appropriate choice for downstream analyses. In short, we took into account results from the cited benchmark studies (see refs. 18-19 and 37-39 cited in the manuscript) to select candidate methods for integration. Harmony performed the best as it most optimally maintained the global and local structure of the data in dimensionally reduced space. Critically, only Harmony was able to reconstitute the continuum relationship between muscle stem/progenitors and mature skeletal myofibers across both assay sources (cells/nuclei). We found that Harmony was the only method that was able to partially correct for the difference between single-cell and single-nuclei data, and avoid unmixed cell populations due to remaining technique biases. This is especially important for skeletal muscle, as both assay methods are critical for analyzing the complete repertoire of cell types within the tissue. Finally, we found that Harmony has low memory requirements, a short run time, and was easiest to add into standard Seurat workflows. Each of these practical points is important for building repeatable and reusable computational pipelines for others to use. See revised text in lines 109-121.

3) UMAPs of scRNAseq show nicely the separation of cell types in Figure1D (and figure S2) after integration with Harmony; however, a control UMAP showing results with no batch effect correction should be included. Similarly, Fig1C should also include a UMAP showing results after integration with batch effect correction.

Response #14: We thank Reviewer #2 for this comment. We clarify that **Fig. 1C** shows the UMAP embedding prior to any batch-correction. We chose not to perform clustering and cell type annotation prior to batch-correction because the batch effects confound these analyses and leave disparate cell clusters solely due to data source and assay method. **Fig. S2** demonstrates the removal of batch-effects after Harmony by showing the mixing of samples in UMAP space and can be compared to **Fig. 1C-D**.

4) Some results are not clearly described: 1) Fig3B: what do the different colors represent for, different data source? Please describe. 2): what does “cultured MuSCs” stand for, in line 163 and Figure 2E?

Response #15: We thank Reviewer #2 for pointing out these errors. We have updated our figure legends and text to correct these issues.

Minor concerns:

- 1) In line 197, Fig.3C should be Fig.3D according to the description.
- 2) Figure S1, labels D) and E) are missing in Fig1SD and Fig1SE.
- 3) Figure S2, labels for all clusters should be provided in all figures, as being provided in the Harmony analysis figures.
- 4) FigureS3, labels A), B) and C) are missing.

Response #16: We thank Reviewer #2 for pointing out these errors. We have updated our figure legends and text to correct these issues.

5) In line 296, "... We created a public web tool (scmuscle.bme.cornell.edu) to enable...", the website link is not working.

Response #17: The web tool (scmuscle.bme.cornell.edu) is operational, though often requires significant (~3-5 minutes) load times in certain browsers. We note that we have added new functionalities to our web tool at including more downloadable data options and added further detail to our bioinformatic pipelines at the Github repository linked in the Methods section (<https://github.com/mckellardw/scMuscle>).

Reviewer #3 (Remarks to the Author):

In the manuscript "Strength in numbers: Large-scale integration of single-cell transcriptomic data captures transitional progenitor states in muscle regeneration", McKellar et al. very efficiently integrate single-cell (scRNAseq) and single-nuclei (snRNAseq) RNAseq data from 23 newly collected and 88 publicly available datasets. The analysis covers two important aspects: it allows to follow the transitional states of the transcriptome during myogenesis (stem cells in a quiescent status, committed progenitors, transitional fusion states to myofiber maturation), it captures the predominant cell types present in skeletal muscle and involved in muscle repair. The authors add the first spatial RNAseq report, performed at different time points after muscle injury. There is no doubt that the pieces of information reported in the manuscript are of value for the muscle field. The comprehensive view of the muscle transcriptome at different stages, and the web tool publicly available will help to enhance our knowledge of muscle biology. As such, the manuscript is a good candidate for publication in Communications Biology. Clarifications of some points will help to prove some of the findings.

Response #18: We thank Reviewer #3 for the appreciation of our work and for their valuable comments and suggestions.

1. The pipeline and methods used to combine different datasets and limit batch effects are well explained and convincing. In reference to Fig S1, it will benefit the readers if the authors can include a more detailed analysis of scRNAseq vs snRNAseq. Do they show a comparable induction of dissociation-induced stress genes? Are the major RNA signatures conserved?

Response #19: We thank Reviewer #3 for this comment. We agree that defining the examining the benefits and biases of integrated single-cell and single-nucleus RNA-seq data, both globally across all cells and in each specific cell type, worthwhile. To this end, we now show in Fig. 1E that a previously identified (Machado et al, Cell Stem Cell, 2021) set of genes associated with dissociation-induced stress ("stress core") are enriched in single-cell data and largely absent in single-nucleus data across all cell types. We further performed differential enrichment analyses

on the stress core gene set and other RNA bio-types as well as individual genes between the two assay methods for each cell type. These analyses are presented in new **Figs. S5** and **S6** and described in detail in Responses #2 and #11 above.

2. The idea of identifying new surface markers to enrich for rare transient cell types is very interesting. It will be valuable to test some of the listed markers in a FACS isolation followed by bulk RNAseq to have a more comprehensive view of the transcriptome of transient committed/fusing cells. Alternatively, combined immunostaining for some of the newly identified surface markers can strengthen the conclusions of their bioinformatic analysis.

Response #20: We appreciate the Reviewer #3's suggestion to validate a subset of suggested surface markers. We found this suggestion to require substantial new experiments and argue that it is beyond the scope of this revision. We posit that the surface receptors identified in **Fig 3D** (for myogenic cells) and in new **Fig. S4** (for FAP, endothelial, and myeloid immune cell sub-populations) provide a potentially valuable resource of candidate surface receptors that can be explored for specificity of expression in future studies.

3. The large-scale integration of single-cell transcriptomic data led to the identification of a core of transcription factors highly expressed during the commitment and fusion stages of myogenesis. Can the authors identify any network of key genes or pathways controlled by the listed transcription factors?

Response #21: We appreciate the Reviewer #3's suggestion to examine gene regulatory networks involved in the transcription factors identified by their expression patterns within the myogenic trajectory (**Fig. 3D**). We concluded that this would require extensive analysis and expand significantly beyond the scope of this manuscript, and therefore prioritized other revisions.

4. In figure 4F the authors show an interesting analysis of the co-occurrence of different cell types within each spot on the spatial transcriptomic slide. Is there any difference in the transcriptome of myogenic cells located close to macrophages, compared to myogenic cells not occupying the same spot? Is there any other cross-influence in the transcriptome between different cell types?

Response #22: We appreciate the Reviewer #3's this suggestion and refer to Responses #4 and #10 above on similar topics.

REVIEWERS' COMMENTS:

Reviewer #1 (Remarks to the Author):

Congratulations on the nice job addressing my previous concerns

Reviewer #2 (Remarks to the Author):

The authors have addressed all my critiques. I have no more concerns.

Reviewer #3 (Remarks to the Author):

The revisions have clarified questions and improved many aspects of the manuscript, addressing most of the initial concerns.

In the present form, the manuscript will be helpful and useful to many in the field.